# Phenolic compounds of *Theobroma cacao* L. show potential against dengue RdRp protease enzyme inhibition by *In-silico* docking, DFT study, MD simulation and MMGBSA calculation

**A. K. M. Moyeenul Huq[1,2☯], Miah Roney[1,3☯], Amit Dubey[4,5], Muhammad Hassan Nasir[6], Aisha Tufail[5], Mohd Fadhlizil Fasihi Mohd Aluwi[1,3], Wan Maznah Wan Ishak[7], Md. Rabiul Islam[8]\*, Saiful Nizam Tajuddin[1,3]\***

1 Centre for Bio-Aromatic Research, Universiti Malaysia Pahang Al Sultan Abdullah, Kuantan, Pahang Darul Makmur, Malaysia, 2 Department of Pharmacy, School of Medicine, University of Asia Pacific, Dhaka, Bangladesh, 3 Faculty of Industrial Sciences and Technology, Universiti Malaysia Pahang Al Sultan Abdullah, Kuantan, Pahang Darul Makmur, Malaysia, 4 Department of Pharmacology, Saveetha Dental College and Hospital, Saveetha Institute of Medical and Technical Sciences, Chennai, Tamil Nadu, India, 5 Department of Computational Chemistry and Drug Discovery Division, Quanta Calculus, Greater Noida, Uttar Pradesh, India, 6 Faculty of Medicine, University Sultan Zainal Abidin (UniSZA), Kuala Terengganu, Terengganu Darul Iman, Malaysia, 7 Faculty of Chemical and Processing Engineering Technology, Universiti Malaysia Pahang Al Sultan Abdullah, Kuantan, Pahang Darul Makmur, Malaysia, 8 School of Pharmacy, BRAC University, Dhaka, Bangladesh

☯ These authors contributed equally to this work.
\* robi.ayaan@gmail.com (MRI); saifulnizam@ump.edu.my (SNT)

**Data Availability Statement:** All relevant data are within the manuscript and its Supporting information files.

## Abstract

### Background

Currently, there is no antiviral medication for dengue, a potentially fatal tropical infectious illness spread by two mosquito species, *Aedes aegypti* and *Aedes albopictus*. The RdRp protease of dengue virus is a potential therapeutic target. This study focused on the in silico drug discovery of RdRp protease inhibitors.

### Methods

To assess the potential inhibitory activity of 29 phenolic acids from Theobroma cacao L. against DENV3-NS5 RdRp, a range of computational methods were employed. These included docking, drug-likeness analysis, ADMET prediction, density functional theory (DFT) calculations, and molecular dynamics (MD) simulations. The aim of these studies was to confirm the stability of the ligand-protein complex and the binding pose identified during the docking experiment.

### Results

Twenty-one compounds were found to have possible inhibitory activities against DENV according to the docking data, and they had a binding affinity of $\geq$-37.417 kcal/mol for

**Funding:** Saiful Nizam Tajuddin received partial funding for this study from Lembaga Koko Malaysia (University Reference Number: RDU 210710)." instead of "The author(s) received no specific funding for this work.

DENV3- enzyme as compared to the reference compound panduratin A. Additionally, the drug-likeness investigation produced four hit compounds that were subjected to ADMET screening to obtain the lead compound, catechin. Based on ELUMO, EHOMO, and band energy gap, the DFT calculations showed strong electronegetivity, favouravle global softness and chemical reactivity with considerable intra-molecular charge transfer between electron-donor to electron-acceptor groups for catechin. The MD simulation result also demonstrated favourable RMSD, RMSF, SASA and H-bonds in at the binding pocket of DENV3-NS5 RdRp for catechin as compared to panduratin A.

## Conclusion

According to the present findings, catechin showed high binding affinity and sufficient drug-like properties with the appropriate ADMET profiles. Moreover, DFT and MD studies further supported the drug-like action of catechin as a potential therapeutic candidate. Therefore, further *in vitro* and *in vivo* research on cocoa and its phytochemical catechin should be taken into consideration to develop as a potential DENV inhibitor.

## Introduction

A virus transmitted by mosquitoes called dengue has recently become widespread around the world. According to Wilder-Smith et al., almost half of the population of the world is susceptible to dengue infection, particularly in the tropics and subtropics [1]. Every year, between 50 and 100 million clinical cases are documented, and more than 500,000 people have dengue shock syndrome (DSS) or dengue hemorrhagic fever (DHF), which are severe signs of dengue infections. As of right present, there is no specific medication available to treat DENV infections. For patients with DENV infections that are hospitalized, supportive therapy is the sole choice for treatment.

DENV is a member of the Flaviviridae family, which also comprises the four serotypes DENV1, DENV2, DENV3, and DENV4 [2]. In DENV-endemic regions, all four serotypes often coexist. The virus infects people when female *Aedes aegypti* or *Aedes albopictus* mosquitoes bite them, resulting in clinical symptoms that can range from mild to severe. One of the fundamental challenges to the development of a DENV vaccine is the documented promotion of severe illness by subsequent infections with serotypes distinct from the original infection [3]. A tetravalent dengue vaccine called CYD-TDV (Dengvaxia, Sanofi Pasteur, Lyon, France) was recently given the go-ahead in 20 endemic nations. The World Health Organiztion (WHO) advises against administering CYD-TDV to seronegative people since it increases the risk of developing severe dengue [4]. CYD-TDV should only be administered to dengue-infected people aged 9 to 45. However, this vaccine's overall effectiveness was said to be restricted to just average protection against DENV1 and DENV2 (protection rates of 50% and 35–42%, respectively) [3]. The discovery of anti-DENV medications in the treatment of DENV infection are crucial for avoiding illness development, lowering disease severity, and halting the transmission of the virus, even if the availability of a DENV vaccination is essential for the prevention and control of viral infection.

The positive-sense RNA genome of DENV is single-stranded and approximately ~11 kb in size. It actively contributes to the production of RNA signals that control the viral replication process [5, 6]. Three structural and seven non-structural proteins surround a single lengthy

open reading frame (ORF) that is the product of the gene. The capsid, membrane, and envelope proteins are structural proteins, while NS1, NS2A, NS2B, NS3, NS4A, NS4B, and NS5 are nonstructural proteins [7]. These nonstructural proteins play a crucial role in the success of viral replication in flaviviruses, and nonstructural protein 5 (NS5) is the biggest and most conserved of them all. The bifunctional DENV-NS5 protein has 900 amino acids [8]. It is a big oligomer that plays important enzymatic tasks such as catalysing 5'-RNA methylation and RNA synthesis, respectively [9, 10]. It has a methyltransferase (MTase) domain at the N-terminal and an RNA-dependent RNA polymerase (RdRp) domain at its C-terminal. These enzymes are located as distinct domains inside a polypeptide and are encoded in a single multifunctional protein. RdRp and other viral polymerases are known therapeutic targets in clinical practice. Additionally, RdRp is a crucial target for anti-dengue drug development since it is the most conserved viral protein across all four DENV serotypes [11], as well as among other flaviviruses, including the West Nile Virus (WNV) and Yellow Fever Virus.

By providing consistent supplies of bioactive lead molecules, natural products play a significant role in the finding of leads for the development of chemical therapies [12]. According to statistics recently released by the US Food and Drug Administration (FDA), natural products made up 67% of the 1562 small molecules that were approved for sale between 1981 and 2014 [13–15]. David et al., have classified these substances as either novel chemical entities (NCEs), unmodified natural products (NPs), biological macromolecules, herbal medicines, NP derivatives, structurally similar NPs, semisynthetic NPs, analogues, or synthetic compounds based on NP pharmacophores [16].

Chocolate and other food products made from the *Theobroma cacao* L bean are very popular across the world among all aged group of people. Pro capita chocolate consumption in Europe ranges from 1.04 kg per year in Poland to 11.85 kg per year in Ireland [17]. Due to its distinctive chemical makeup of more than 500 different chemicals, cocoa has emerged as a valuable ethnomedicinal plant [18]. The following pharmacological activities have been linked to their reported benefits for human health: analgesic [19], immunomodulatory [20], vasodilatory [21], antioxidant [22], anti-inflammatory, anticarcinogenic [23], immunomodulatory [20], vasodilatory [21], analgesic [19], antimicrobial [24] and antiviral [25, 26] activities. Cocoa has long been regarded as a food that is high in polyphenols. The kind of polyphenols is mostly formed by flavonoids and phenolic acids. Indeed, phenolic acids have been shown to have a wide range of biological effects, particularly those that are anti-inflammatory [27], anti-malarial [28], antioxidant [29], anticancer [30], and antiviral [31] effects.

Computational drug design (CADD) employs theoretical concepts and computer methods to investigate potential drug interactions with biomolecules [32, 33] involving a number of techniques and tools such as molecular docking, molecular dynamic simulation, ADMET, and drug-likeness. Molecular docking, a key aspect of CADD, is commonly utilized to forecast the binding patterns of known ligands, discover new potent ligands, and estimate binding affinities with target proteins. It aids in lead or hit identification based on chemical diversity, known biological activity, or potential for drug development [34]. Overall, molecular docking signify the drug discovery process, enabling the understanding of interactions between a ligand and its target, crucial for designing new drugs, optimizing existing ones, and uncovering molecular mechanisms in biological processes [32]. Additionally, molecular dynamics (MD) simulations are pivotal in drug discovery, offering insights into the atomic-level behavior of biological systems. MD simulations have been applied for pharmacophore development, drug design, and identifying compounds that complement the receptor [35]. ADMET and drug-likeness approaches have gained importance due to the abundance of ligand property data and the creation of on-demand virtual libraries of drug-like small molecules [36]. Virtual screening, exemplified by molecular docking, presents an affordable,

rapid, direct, and systematic approach to drug discovery compared to traditional experimental high-throughput screening [37].

About 29 phenolic compounds from *Theobroma cacao* L. were docked to the DENV-3 NS5 RdRp in this work. The top hits underwent additional *in-silico* testing for druggability, pharmacokinetics, and pharmacodynamics characteristics. The lead chemical was subsequently put through a DFT and MD simulation analysis to check the accuracy and stability of the docking.

## Methodology

### Protein selection and preparation

The most crucial phase in the drug discovery process is choosing a prospective therapeutic target or receptor that is involved in dengue replication. To find a dengue virus inhibitor, the target protein of DENV-3 NS5 RdRp was chosen with the PDB ID: 6IZZ [38]. The Protein Data Bank (http://rcsb.org) was used to retrieve the target protein's crystal structure.

Discovery Studio 3.1 and Chimaera 1.5.3 were used to prepare the protein. Hydrogen atoms were inserted where amino acid residues were missing them, loop segments were added around the active regions of macromolecules, and various bond configurations were checked again. The PDB files also no longer contain any crystallographic waters [39].

### Ligand selection and preparation

ChemDraw was used to generate a two-dimensional (2D) structure of 29 phenolic compounds that were extracted from *Theobroma cacao* L., and the structures that were taken from Yaez et al., [40].

With the help of Discovery Studio 3.1 and the CHARMm force field, the three-dimensional (3D) structure was generated from a 2D model. To create low-energy ring conformations, the CHARMm force field in DS3.1 was used to produce the ligand. By default, all compounds were adjusted to a pH range of 5.0 to 9.0 for an appropriate protonation state.

### Molecular docking validation

To validate the molecular docking process, the co-crystallized ligand was redocked into the binding site of the target protein using DS3.1. The prior docking position of compounds and the lowest energy posture achieved after re-docking were overlaid, and its root mean square deviation (RMSD) was determined. The RMSD must be contained within the 2.0 Å reliable range in order to certify the docking operation.

### Molecular docking

Utilising the DS3.1, molecular docking was performed on the Theobroma cacao L. phenolic compounds and the DENV-3 NS5 RdRp crystal structures (PDB ID: 6IZZ; 1.97). Under the CDOCKER technique, rigid docking was used to anticipate the interaction between a ligand and a protein. A grid box made up of the dimensions -15.65 Å × -19.54 Å ×38.24 Å was created around the active site using the DS3.1 software using protein PDBQT input files, and docking calculations were carried out inside an 8.24 Å sphere. The co-crystallized ligand's binding position determined the site sphere centre, and all other parameters were kept at their normal settings. The exclusion was carried out using the Root Mean Square Deviation (RMSD) parameter in accordance with the co-crystallized compound, with the radius set at 10. Assuming that each ligand's best 10 conformations were preserved based

on scoring and ranking by the negative value of CDOCKER capacity, the best Hits parameter was set to 10.

Additionally, in the most recent induced fit docking simulation, molecules that are being examined as potential ligands must have lower binding interaction energies than the reference ligands. In this investigation, panduratin A served as the reference compound. Panduratin A is a phenolic derivative of a natural substance that has an anti-DENV protease with $IC_{50}$ value of 57.2871.30 μmol/L [41]. Additionally, with an $IC_{50}$ value of 0.81 μM against SARS-CoV-2, this substance also demonstrated anti-SARS-CoV-2 action [42].

## Physicochemical properties and drug-likeness study

Through the online tool SwissADME (http://www.swissadme.ch), the molecular characteristics and applicability of the compounds as drug candidates were assessed based on the "Lipinski's Rule of Five" cutoff point. Molecular weight, molar refractivity, topological polar surface area, the number of O or N, the number of OH or NH, and the number of rotatable bonds are the six descriptors according to Lipinski's rule. The threshold some method, such as Lipinski, Ghose, Veber, Egan, and Muegge, as well as the bioavailability score of a molecule, were used to determine the drug-likeness research.

## ADMET study

In order to ascertain the pharmacokinetic characteristics of the compounds, ADMET prediction was used to forecast the hit compounds from the Physicochemical Properties and Drug-likeness Study. The pkCSM web server (http://biosig.unimelb.edu.au/pkcsm/prediction) was used to conduct these experiments. For the preparation, just SMILES or SD file structures of active compounds are needed; understanding of the active site or binding mechanism is not required. As typical predictors, we looked at aqueous solubility, human intestinal absorption, blood-brain barrier, CYP450 substrate and inhibitor, hERG I and II inhibitor, hepatotoxicity, and skin sensitization.

## Density function theory (DFT) calculations and the molecular electrostatic potential (MESP) calculations

The electronic attributes of the compounds were determined using quantum mechanics (QM) theories. The computational chemistry calculations in this study were based on one of the most effective methods for analyzing compound stability and reactivity. Vibrational frequency calculations using density functional theory (DFT) are critical for theoretical studies of organic molecules and related domains [43, 44]. Furthermore, the DFT technique is an important and useful tool for investigating the relationship between the geometry and electrical properties of bioactive molecules [44–46]. As a result, we present DFT calculations involving HOMO--LUMO energy, various chemical reactivity parameters, and the electrostatic potential of a molecule. Using the HOMO and LUMO values, numerous other computations, such as chemical hardness, softness, electronegativity, and electrophilicity, have been performed. The compound's complete molecular geometry optimization was performed using the Gaussian 09 software package and the density functional theory at the B3LYP/6-31G (d, p) level of theory [47]. Using their optimized structures, the molecular electrostatic potential, highest occupied molecular orbital, and lowest unoccupied molecular orbital energies of compounds catechin and panduratin A (control) were calculated.

## Molecular dynamic simulations of the top-scoring molecule with anti-dengue RdRp protease

The GROMACS 2023.2 software was employed to perform molecular dynamics (MD) simulations on the protein-panduratin and protein-catechin complexes, as well as the apo-protein. The CHARMM27 force field and the pre-configured TIP3P solvent model within the GROMACS module were utilized to generate topologies for the protein-ligand complexes under investigation. Subsequently, these test systems were positioned within a cubic water box featuring a 12-surface polyhedron as its edge, set at a distance of 1.0 nm. Neutralization of the system was achieved by introducing appropriate Na+ and Cl- ions. The protein-panduratin A and protein-catechin complexes underwent an initial energy minimization for 100 fs under "isothermal-isobaric" conditions, followed by equilibration in the NVT and NPT phases. Data were collected throughout a 100 ns simulation using a 2fs time step. To assess the stability of the complexes, measurements of the root mean square deviation (RMSD) for each C$\alpha$ atom, RMSF, SASA, and hydrogen bonds (HB) were recorded at 100 ns intervals during the MD trajectories.

## Binding free energy calculations of top-scoring molecule with anti-dengue RdRp protease

The protein-ligand interaction binding free energies, including van der Waals, electrostatic, polar salvation, SASA, and binding energies, may be integrated with high-throughput MD simulations using MM-PBSA calculations. Here, we employed the g_mmpbsa package byapplying the single step computing method. Here we executed g_mmpbsa with the protein-ligand MD trajectory data and parameter file. With the help of the MmPbsaStat.py Python script, which is offered by the g_mmpbsa package van der Waals, binding, electrostatic, and polar salvation energies are computed. Utilising MmPbSaDecomp.py from the g_mmpbsa package allowed us to also forecast the contribution energy of each residue.

## Ethics

Ethical approval was not required for this work.

## Results and discussion

### Molecular docking validation

The co-crystallized ligand of the target protein was re-docked into each of its individual binding sites on the DENV-3 NS5 RdRp (PDB ID: 6IZZ; resolution: 1.97 Å) in order to confirm the docking procedure. Re-docked co-crystallized ligand had a root mean square deviation (RMSD) of 0.39 Å, which was less than 2.0 Å. In Fig 1, the docked structure and co-crystallized ligand are superimposed.

### Molecular docking

In this work, 29 phenolic compounds were extracted from *Theobroma cacao* L., and docked against DENV-3 NS5 RdRp enzyme. The top-ranked compounds were determined by comparing them to the reference compound apnduratin A and determining which had the lowest docking energy value and the greatest number of contacts with active site residues. Potential inhibitors can be thought of as substances that can thwart virus-host binding and prevent viral particles from entering target cells. The amino acid residues with the highest binding affinity

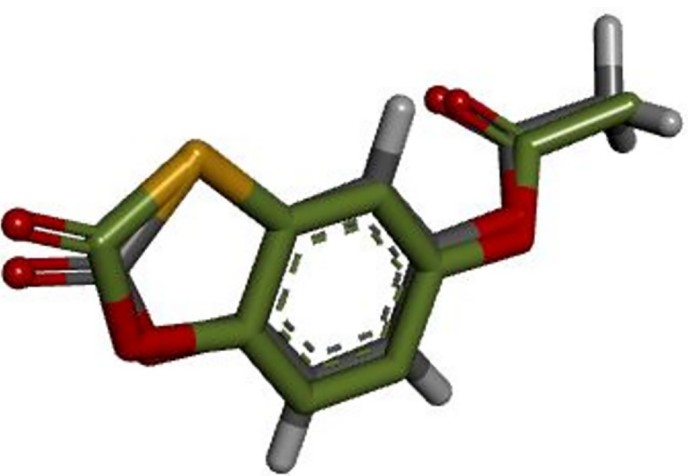

**Fig 1. Superimposed of the crystal structure (green) and docked structure (dark gray).**

and interactions with the reference chemical were therefore chosen for additional study. The interactions were, however, favourable, as indicated by the negative docking energy values.

The suppression of viral replication depends on NS5 RdRp [41, 48, 49]. Compared to the reference molecule panduratin A, 21 compounds exhibited the lowest interaction energies, the highest number of hydrogen bonds, and the most significant hydrophobic interactions upon docking at the binding site of DENV-3 NS5 RdRp (S1 Table). In contrast to panduratin A, which displayed a docking energy (CDOCKER interaction energy) of -37.4166 kcal/mol, the selected compounds demonstrated a range of -39.287 to -60.4769 kcal/mol. The stability of the ligand-protein complexes was primarily attributed to hydrogen bonds and carbon-hydrogen bonds formed with the residues of the active site. The study unveiled additional stabilizing factors such as π-sulphur interactions, π-π stacked interactions, and π-alkyl interactions at the binding site of the target protein. These interactions further contributed to the stability of the complexes. The active site residue, common to all 21 compounds, was discussed earlier for its pivotal role in maintaining the structural and functional integrity of NS5 dengue proteases. The docking poses for panduratin A and catechin are illustrated in Fig 2.

## Physicochemical properties and drug-likeness study

The SwissADME server's drug scan algorithms forecast the potential drug-likeness of the suggested DENV inhibitors. The Lipinski rule of five was used to filter all of the chosen compounds in order to determine their physicochemical characteristics. The drug-like molecules

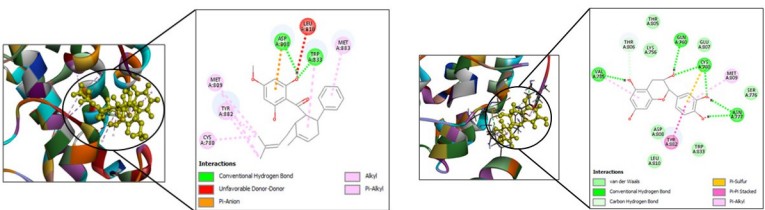

**Fig 2. Docking poses for (a) Panduratin A and (b) Catechin in PDB: 6IZZ.**

were identified based on the threshold of some methods including Lipinski, Ghose, Veber, Egan, and Muegge as well as the bioavailability score of a compound. The six descriptors included molecular weight, molar refractivity, topological polar surface area, number of O or N, number of OH or NH, and number of rotatable bonds. Only four of the tested compounds —catechin, isorhamnetin, luteolin, and quercetin—passed all drug-likeness assays with flying colours (S2 Table). The substances that successfully satisfied all of these criteria were identified as possible lead substances with good pharmacokinetic properties and were then the focus of additional ADMET research.

## ADMET study

Additional ADMET calculations were performed to forecast the inhibitor's capacity to mimic a medication. The criteria included in this investigation were toxicity, absorption, distribution, and metabolism. The number of thresholds has then been used to assess these parameters. The findings indicated that all of the chosen compounds did not meet the requirements for some metrics. However, the catechin inhibitor's ADME prediction result indicated that it would be able to surpass the ADMET drug ability threshold (S3 Table).

The ADMET prediction result for catechin indicated some water solubility (Log S) with a value of -3.117 and moderate absorption in the human gut with a value of 68.829. The substance failed to cross the blood-brain barrier and serve as a non-substrate of P-glycoprotein. However, it was determined that this substance did not inhibit the CYP1A2, CYP23A4, CYP2C9, or CYP2D6 subtypes of the enzyme. The results of this *in silico* ADMET study also indicated that Catechin would not be harmful to hERG I or II, cause hepatotoxicity, or cause skin sensitization. Due to the fact that there were no breaches of Lipinski's rule of five, and because ADMET analysis anticipated catechin to be a possible inhibitor, the findings of Lipinski's calculation and ADMET analysis showed that it was the most preferred among the others.

## Density function theory (DFT) calculations

The values of HOMO and LUMO energy can be used to define a molecule's ability to donate and receive electrons. These molecular orbitals are important for electronic and optical properties, pharmaceutical research, and understanding biological mechanisms [50, 51]. The energy gap supports of the frontier molecular orbital (FMO) structure indicate the structure's stability. Furthermore, FMOs provide information about a molecule's kinetic stability and chemical reactivity. The energy values determined for catechin's HOMO and LUMO orbitals are -0.2195 eV and -0.0053 eV, respectively. The FMO molecule's energy gap (HOMO--LUMO) was determined to be 0.21412eV. The lower HOMO and LUMO energy gaps indicated that the investigated molecule has strong chemical reactivity, biological properties, and polarizability (Figs 3 and 4). A large energy gap generally indicates a hard and stable molecule, whereas a small energy gap indicates a soft and reactive molecule. Catechin has a lower energy gap of 0.21412 eV, making it a soft, reactive, and polarizable molecule. Furthermore, chemical reactivity parameters such as chemical softness (S), chemical potential (m), electrophilicity index (u), and chemical hardness (h) of the studied molecules were calculated using the energies of the HOMO and LUMO orbitals. Because of the molecule's small energy gap (0.21412 eV), significant softness (9.3 eV), and low chemical hardness (0.1 eV), catechin is a promising candidate for use as a chelating agent. The electronegativity of the control molecule is slightly greater than that of the catechin, indicating that the control molecule is more stable and the catechin is slightly more reactive. The electrophilicity index of catechin indicates its ability to bind to biomolecules [52–54]. The molecule of interest (catechin) has nearly identical values to that of control molecule panduratin A. Table 1 contains a list of all DFT descriptors.

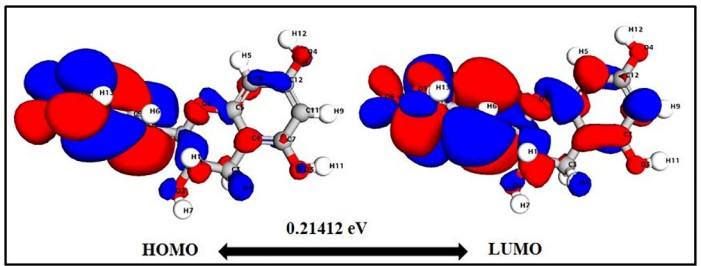

**Fig 3. Catechin HOMO-LUMO map with energy gap.**

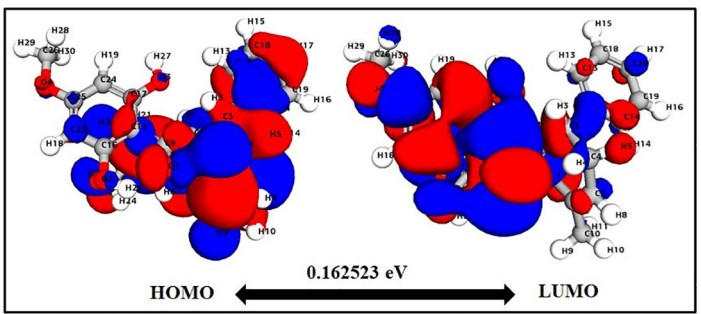

**Fig 4. Panduratin A HOMO-LUMO map with energy gap.**

## The molecular electrostatic potential (MESP) calculations

The molecular electrostatic potential (MEP) predicts the relative reactivity positions of a species for nucleophilic and electrophilic attack. The significance of MESP lines demonstrates the color marking system's design, size, negative, positive, and neutral electrostatic potential zones. The MEP surface analysis of the compounds were determined using the optimized structures and the B3LYP/6-31G (d, p) basis set. Figs 5 and 6 shows the electrostatic potential surface map of the studied compound as well as the control. The blue color scheme area represents the significant positive electrostatic potential of the compounds (indicating a significantly electron-deficient region), whereas the red color area represents the compounds' most

**Table 1. Showing DFT descriptors of the compound and control.**

| Name | Catechin | Panduratin A |
|---|---|---|
| Total energy | -1092.14 | -1389.03 |
| Binding energy | -70.231 | -94.257 |
| Dipole moment | 1.54211 | 3.53376 |
| HOMO energy | -0.219516 | -0.216736 |
| LUMO energy | -0.00539583 | -0.0542137 |
| Band Gap Energy | 0.21412 | 0.162523 |
| Hardness | 0.10706 | 0.0812615 |
| Softness | 9.340556697179151 | 12.3059505423847701556 |
| Electronegativity | 0.112455915 | 0.13547485 |
| Electrophilicity | 0.0590618943512386 | 0.1129282315888982 |

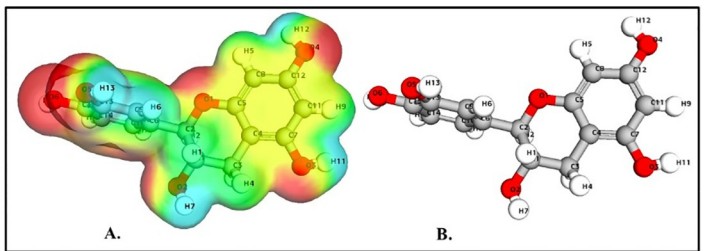

**Fig 5. (A) Molecular electrostatic potential of catechin (B) Optimized structure of catechin.**

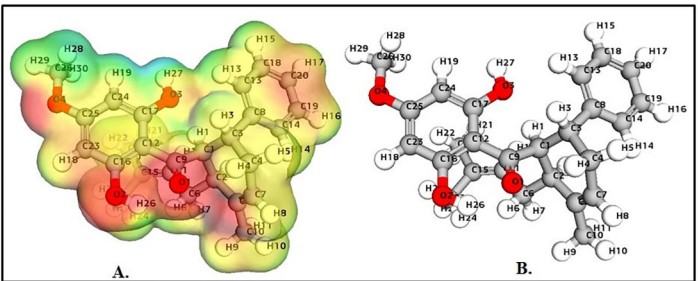

**Fig 6. (A) Molecular electrostatic potential of panduratin A (B) Optimized structure of panduratin A.**

electronegative potential (indicating an electron-rich region). The negative potential regions in the MESP are localized over electronegative atoms (oxygen and carbon), while the positive electrical regions are localized over hydrogen atoms. As a result, nucleophilic and electrophilic species prefer sites with higher negative electronegative potential and lower positive electrostatic potential. Electronic charges play a significant role in describing the bonding abilities of a compound.

In the case of catechin, highly negative points were localized at atoms O1, O2, O3, O4, O5, and O6 whereas the highly positive points were located at atoms H1, H4, H6, H7, H11, H12, and H13 (Fig 5). Whereas in the case of panduratin A, the electrostatic map reveals that highly negative points were located at atoms C2, C6, C9, C15, H6, H7, O1, O2, and O4 atoms, whereas the highly positive points are present at H19, H27, H28, and H29 atoms (Fig 6). The degree of negative Hirshfeld charge on the targeted molecule, displaying nucleophilicity, strongly correlates with its ability to donate electrons to the approaching electrophile. S4 Table shows the mulliken charge values of the examined compound's component atoms. The computed bond properties are shown in S5 Table.

The comparison between catechin and panduratin A highlights distinct reactivity and interaction mechanisms based on various molecular properties. Panduratin A exhibits considerably higher total energy and binding energy in comparison to catechin, indicating slightly lower stability and a potentiality for forming robust interactions during reactions or when binding to other molecules as compared to panduratin A. Moreover, panduratin A possesses a substantially larger dipole moment, signifying higher polarity and a heightened potential for strong electrostatic interactions. While both compounds share similar HOMO energies, catechin showcases a slightly higher LUMO energy and an extended band energy gap, hinting at a slightly lower reactivity potential due to its lesser likelihood to donate or accept electrons. Additionally, catechin's increased hardness and lower softness compared to panduratin A

suggest a more controlled reactivity, indicating a lesser susceptibility to changes in electron density. The compound's higher electronegativity and electrophilicity further reinforce its potential for increased reactivity and stronger interactions in various chemical contexts. It is crucial to note that these observations offer a molecular perspective, and the actual reactivity and interaction mechanisms can be influenced by external factors and specific experimental conditions.

Understanding the molecular properties of compounds such as catechin and panduratin A is critical when exploring their potential inhibitory action on the RdRp (RNA-dependent RNA polymerase) enzyme in diseases like dengue, where RdRp plays a pivotal role in the virus's replication. Panduratin A's traits, like higher stability indicated by elevated total and binding energies, suggest its capacity for establishing robust interactions with the RdRp enzyme. The larger dipole moment and increased polarity might enable panduratin A to engage in specific interactions with crucial regions of the RdRp protein, potentially influencing its binding capacity and inhibitory potential. Comparable and similar properties of catechin may also show favorable inhibitory action of RdRp. Additionally, the lower LUMO energy and smaller band gap energy of compounds hint at a greater likelihood for electron transfer, advantageous in disrupting electron transfer processes or active sites of the RdRp enzyme, hindering its function and impeding viral replication. Higher hardness and lower softness imply controlled reactivity, beneficial for stable and specific interactions with active sites or critical residues within the RdRp enzyme, potentially interfering with its catalytic activity and inhibiting viral replication. Moreover, compound's higher electronegativity and electrophilicity suggest its capability to act as an electron acceptor, forming robust interactions with specific functional groups or residues within the RdRp enzyme, possibly disrupting its function and contributing to inhibitory action against the dengue virus. Despite the fact that, catechin showed slightly lower outcomes in DFT result in contrast to panduran A, its potentiality to interact with the RdRp can not be denied. While these molecular properties provide insights into both panduratin A and catechin's potential inhibitory action on the RdRp enzyme in dengue virus replication, rigorous experimental investigations, including enzymatic assays, and *in vitro* studies, are necessary to validate these interactions and elucidate specific mechanisms by which catechin interacts with the RdRp protease, thus determining its effectiveness as a potential therapeutic agent against dengue virus infections.

The molecular properties exhibited by compounds like catechin play a critical role in determining their potential as inhibitors, particularly concerning their interaction with specific enzymes like the RdRp protease. These properties contribute to its function as a potential inhibitor in several ways: i) Stability and Binding Energy: A high total and binding energies suggest that catechin could form more stable interactions with the active sites or binding pockets of the RdRp enzyme. This stability is crucial for a strong and long-lasting inhibitor-enzyme interaction, potentially disrupting the enzyme's function necessary for viral replication; ii) Dipole moment and polarity: Larger dipole moment and increased polarity allow it to interact with the RdRp enzyme via electrostatic forces. These interactions may facilitate the compound's binding to specific regions of the enzyme, enhancing its inhibitory potential by disrupting the enzyme's active sites or altering its conformation; iii) HOMO and LUMO energy, band gap energy: Lower LUMO energy and smaller band gap energy suggest its propensity for electron transfer. This characteristic is beneficial as it may interfere with the electron transfer processes essential for the RdRp enzyme's catalytic activity, thereby inhibiting the enzyme's function and viral replication; iv). Hardness and softness: Higher hardness and lower softness imply controlled reactivity. This controlled reactivity could enable the compound to form stable and specific interactions with the active sites or critical residues within the RdRp enzyme, effectively disrupting its function and inhibiting viral replication; v) Electronegativity and

electrophilicity: Higher electronegativity and electrophilicity indicate potential to accept electrons and form strong interactions with specific functional groups or residues within the RdRp enzyme. These interactions could hinder the enzyme's function and contribute to its inhibition.

Collectively, these molecular properties of catechin which is comparable with panduratin A in this study may contribute to its potential as an inhibitor by facilitating strong and specific interactions with the RdRp enzyme. By binding to critical sites or interfering with essential processes in the enzyme's function, catechin in a similar fashion to panduratin A may disrupt the replication cycle of the virus, making it a promising candidate for further exploration as a potential therapeutic agent against viral infections, such as dengue.

## Molecular dynamic simulations of the top-scoring molecule with anti-dengue RdRp protease

We used molecular dynamic simulation to study target receptor and lead compound interactions in the dynamic behavior of both receptor and ligand in order to investigate the stability of bound conformation following binding of lead compound within the binding site of RdRp protease, as molecular docking studies were conducted using the protein rigid crystal structure.

The conformational stability of biological molecules is investigated using the root mean square deviation (RMSD) [55]. RMSD analysis was employed to evaluate the structural modifications induced by catechin and panduratin A (reference) on 6IZZ, and the results are depicted in Fig 7. The overall stability and structural convergence of the two complexes, 6IZZ-catechin and 6IZZ-panduratin A (Reference compound), along with the native system, were scrutinized over a 100 ns simulation period. The 6IZZ-panduratin A complex exhibited sustained stability with minimal fluctuations until 40 ns, after which it experienced a maximum deviation of up to 10.42 nm. Subsequently, it displayed inconsistent fluctuations until 80 ns, followed by a relative stabilization with a mean RMSD of 3.22±2.88 nm, reaching 6.2 nm at the simulation's conclusion.

In contrast, the 6IZZ-catechin complex demonstrated remarkable stability throughout the simulation, with a mean RMSD of 0.32±0.06 nm and a maximum fluctuation of 0.63 nm. Moreover, the average RMSD values of the apo protein (Cα) were 0.21±0.02 nm, suggesting that both complexes were stable, and the deviations were essentially comparable in both the apo form and the catechin-protein complex.

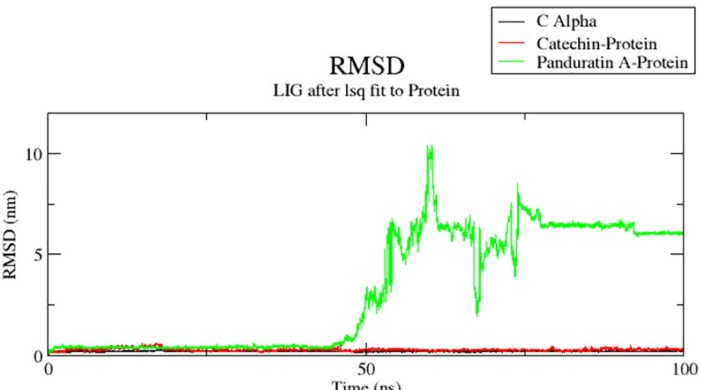

**Fig 7. RMSD of apo protein, protein-catechin, protein-panduratin A complexes.**

Protein residue flexibility is elucidated through the root mean square fluctuation (RMSF) analysis, examining the positional variations of each protein residue in its native state and when bound to the respective ligands [56]. In the 6IZZ-panduratin A complex, the calculated RMSF was 0.17±0.27 nm, while for the 6IZZ-Catechin complex, it was 0.14±0.07 nm. In contrast, the apo-protein displayed average fluctuations of 0.10±0.05 nm (refer to Fig 8). This result suggests that the binding of both compounds slightly increased the mean RMSF value, indicating that the complexes were slightly more flexible upon ligand binding [57]. The analysis reveals that specific residues at the binding site, namely Ala763, Arg773, Ser776, Asn 777, Cys780, Met809, Leu810, Trp833, Leu 880, Tyr882, and Met883, exhibited reduced fluctuation in the presence of panduratin A. Notably, chemical interactions involving Cys780, Met809, Leu810, Trp833, and Tyr882 were also observed in the corresponding docked complexes, suggesting sustained stability throughout the simulation.

Similarly, for catechin, the fluctuations in binding site residues, including Ala759, Gln760, Ala763, Ser776, Asn777, Cys780, Pro784, Val785, Trp787, Pro789, Thr806, Glu807, Asp808, Met809, Trp833, Leu880, and Tyr882, were less pronounced, indicating stability during the simulation. Intriguingly, chemical interactions involving Gln760, Asn777, Cys780, Thr806, Met809, and Tyr882 were consistently present in the corresponding docked complexes, affirming the sustained stability of the docked complex throughout the simulation period.

To comprehend how biomolecules interact, geometric analysis of hydrogen bonding is used. Hydrogen bonding is one of the essential interactions that biomolecules use to preserve their structural integrity [55]. Fig 9a and 9b depict the interactions between the apo-protein and panduratin A, as well as catechin, examined in this investigation over a 100 ns molecular dynamics (MD) simulation. In Fig 9a, the 6IZZ-anduratin A complex reveals the presence of six hydrogen bonds, although they were not consistently maintained throughout the entire simulation. On the other hand, Fig 9b illustrates the 6IZZ-catechin complex, which displays seven hydrogen bonds, with nearly half of them persisting throughout the simulation. This suggests a more robust interaction of catechin with the protein, characterized by a higher number of hydrogen bonds contributing to the stability of the complex.

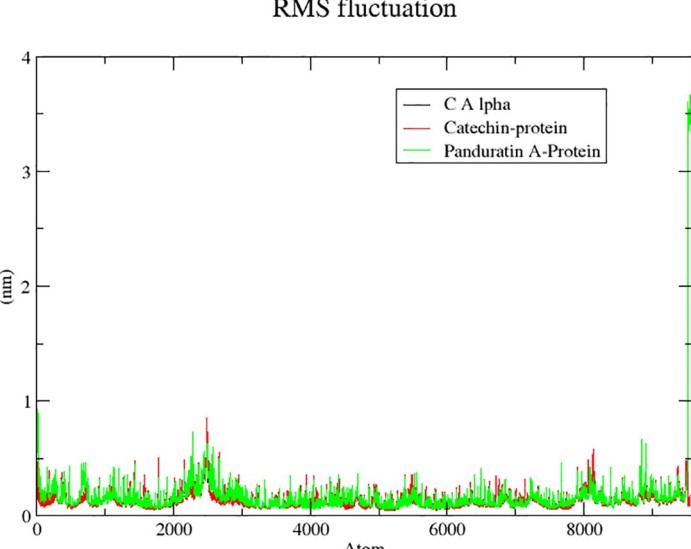

**Fig 8. Root mean square fluctuations of apo protein, protein-catechin, protein-panduratin A complexes.**

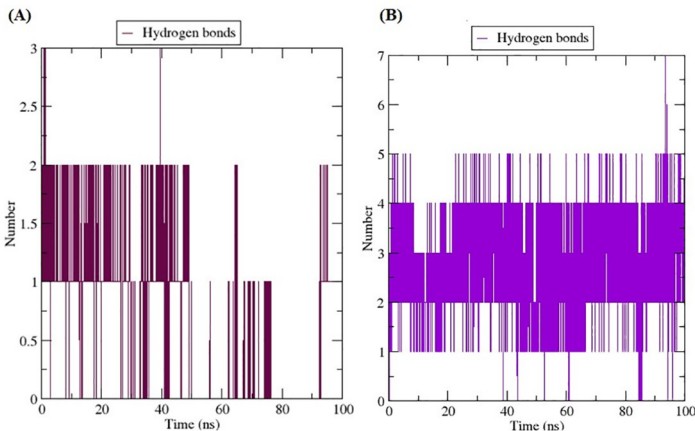

**Fig 9. Number of Hydrogen bonds of (a) Panduratin A and (b) Catechin.**

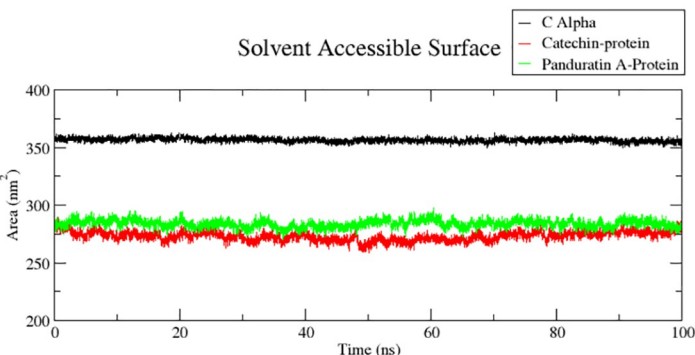

**Fig 10. Solvent-accessible surface area (SASA) analysis of 6IZZ-panduratin A and 6IZZ-catechin complexes and apo-protein.**

To understand the interactions between residues and the surrounding solvent, the polar and non-polar molecular surface areas are calculated using the solvent-accessible surface area (SASA) method [58]. The calculated mean SASA was 356.53±1.74, 283.78±3.48, and 273.14 ±4.17 nm$^2$ for the apo-protein, 6IZZ-panduratin A, and 6IZZ-catchin complexes, respectively (Fig 10). The conformational changes brought about by the both the complexes are evident by the fact that these complexes have lower SASA values than the apo-protein.

## Binding free energy calculations of top-scoring molecule with anti-dengue RdRp protease

The MM-GBSA approach was undertaken to calculate the binding free energies of the 6IZZ-panduratin A and 6IZZ-catchin complexes in order to confirm our investigated results. By taking into consideration all energy sources, the computed free energy sheds light on the ligand binding process. Table 2 displays the outcomes of the MM-GBSA computations for panduratin A and catechin. It is demonstrates that van der Waals forces were mostly responsible for the binding of both compounds. The electrostatic and hydrophobic interactions have major impact on the Gibbs binding free energy. Again, Table 2 unequivocally demonstrates

**Table 2. Gibbs binding free energy (kcal/mol) of 6IZZ-panduratin A and 6IZZ-catechin complexes calculated by MM/GBSA.**

| MM/GBSA (kcal/mol) | Panduratin A | Catechin |
|:---:|:---:|:---:|
| ΔEvdw | -20.13 | -26.16 |
| ΔEelec | -14.40 | -42.93 |
| ΔEpol | 20.22 | 47.54 |
| ΔGnonpol | -2.84 | -4.35 |
| ΔEDISPER | 0.00 | 0.00 |
| ΔGgas | -34.53 | -69.09 |
| ΔGsolv | 17.38 | 43.18 |
| ΔGbinding | -17.15 | -25.91 |

that panduratin A and catechin had total binding free energies of -17.15 kcal/mol and -25.91 kcal/mol, respectively (Fig 11a and 11b, respectively). It was anticipated that the non-covalent binding of both compounds were advantageous for the five parallel groups of ΔEvdw, ΔEelec, ΔGnonpol, ΔGgas, and ΔGbinding. Understanding the role of each residue to ΔG binding will benefit from further deconstruction of the Gibbs binding free energy. Inconsistent from sample to sample is the contribution of residues to ΔGbinding. In order to generate stable connections with panduratin A, the amino acids Arg763, Arg773, Ser776, Asn777, Cys780, Met809, Leu810, Trp833, Leu880, Tyr882 and Met883 provided the negative Gibbs binding energy (Fig 12a). Additionally, in the simulated system, Ala759, Gln760, Ala763, Ser776, Asn777, Cys780, Pro784, Val785, Trp787, Pro789, Thr806, Glu807, Asp808, Met809, Trp833, Leu880 and Tyr882 contributed to the negative Gibbs binding free energy and created stable contacts with catechin (Fig 12b).

## Conclusion

The study of 29 phenolic compounds from *Theobroma cacao* L. revealed that catechin had high binding affinity with the active binding site of DENV3-NS5 RdRp protease. Further investigation on the catechin and target protein interaction through MD simulation for 100 ns period validates the docking study where catechin bound at the active site residues. The

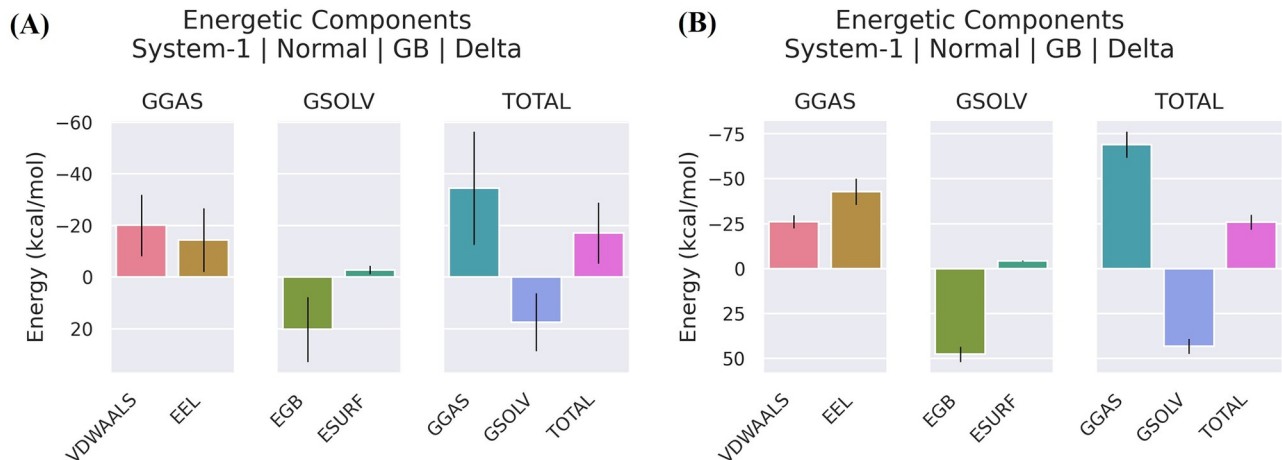

**Fig 11. The Gibbs binding free energy decomposition diagram of (a) 6IZZ-panduratin A and (b) 6IZZ-catechin complexes based on (MM-GBSA).**

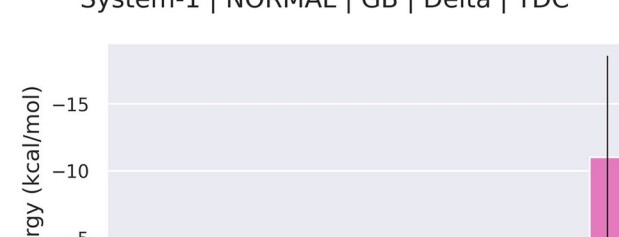
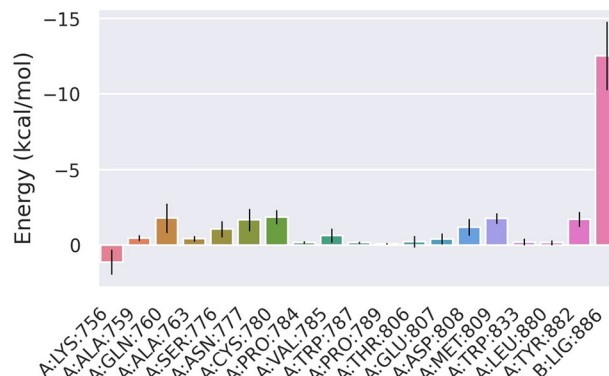

**Fig 12. The Gibbs binding interactions decomposition diagram of (a) 6IZZ-panduratin A and (b) 6IZZ-catechin complexes based on (MM-GBSA).**

reactivity of this molecule was further supported by the DFT study. Moreover, a robust drug-like property with the favourable ADMET profiles of catechin was also demonstrated. This suggests that catechin could be a potential lead or drug candidate. To further design and develop catechin as RdRp protease inhibitor to treat dengue infection *in vitro*, *in vivo* and clinical studies should be undertaken to validate this effectiveness, mechanism of action and toxicity of this compound. Nonetheless, the Cocoa could also be studied further as a potential anti-dengue herbal therapy.

## Supporting information

**S1 Table. Results of the phenolic compounds of *Theobroma cacao* L. against DENV-3 NS5 RdRp protein with their respective docking energy value and interacting residue in the binding site.**
(DOCX)

**S2 Table. Molecular physicochemical descriptors and drug-likeness analysis of the selected compounds.**
(DOCX)

**S3 Table. ADMET profiling enlisting absorption, distribution, metabolism and toxicity related drug-likeness parameters.**
(DOCX)

**S4 Table. Showing properties of atoms of the compound and control.**
(DOCX)

**S5 Table. Showing properties of bonds of the compound and control.**
(DOCX)

## Acknowledgments

We deeply acknowledge the contribution of Dr. Md. Nazim Uddin from Bangladesh Council for Industrial Research (BCSIR), Dhaka for his performing MD simulation in GROMACS.

## Author Contributions

**Conceptualization:** A. K. M. Moyeenul Huq.

**Data curation:** Miah Roney.

**Formal analysis:** A. K. M. Moyeenul Huq, Miah Roney, Amit Dubey, Muhammad Hassan Nasir.

**Funding acquisition:** Saiful Nizam Tajuddin.

**Investigation:** A. K. M. Moyeenul Huq, Miah Roney, Amit Dubey, Muhammad Hassan Nasir, Aisha Tufail.

**Methodology:** A. K. M. Moyeenul Huq, Miah Roney, Mohd Fadhlizil Fasihi Mohd Aluwi.

**Project administration:** Wan Maznah Wan Ishak, Md. Rabiul Islam.

**Resources:** Amit Dubey, Muhammad Hassan Nasir, Mohd Fadhlizil Fasihi Mohd Aluwi.

**Software:** Amit Dubey, Muhammad Hassan Nasir, Mohd Fadhlizil Fasihi Mohd Aluwi.

**Supervision:** Md. Rabiul Islam, Saiful Nizam Tajuddin.

**Validation:** A. K. M. Moyeenul Huq, Miah Roney, Aisha Tufail.

**Visualization:** A. K. M. Moyeenul Huq, Miah Roney.

**Writing – original draft:** A. K. M. Moyeenul Huq, Miah Roney.

**Writing – review & editing:** Wan Maznah Wan Ishak, Md. Rabiul Islam, Saiful Nizam Tajuddin.

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
