## [Decision Letter · Decision Letter 0]

7 Dec 2023

PONE-D-23-34561Phenolic compounds of Theobroma cacao L. showed inhibitory effects on dengue RdRp protease enzyme: Findings from in-silico docking, MD simulation, MMGBSA and DFT calculationsPLOS ONE

Dear Dr. Islam,

Thank you for submitting your manuscript to PLOS ONE. After careful consideration, we feel that it has merit but does not fully meet PLOS ONE’s publication criteria as it currently stands. Therefore, we invite you to submit a revised version of the manuscript that addresses the points raised during the review process.

We look forward to receiving your revised manuscript.

Kind regards,

Erman Salih Istifli, PhD

Academic Editor

PLOS ONE

Journal Requirements:

"This study was supported by the Lembaga Koko Malaysia (University Reference Number: RDU 

210710) to Saiful Nizam Tajuddin."

"The author(s) received no specific funding for this work"

Reviewers' comments:

Reviewer's Responses to Questions

**Comments to the Author**

1. Is the manuscript technically sound, and do the data support the conclusions?

Reviewer #1: Partly

Reviewer #2: Partly

2. Has the statistical analysis been performed appropriately and rigorously? 

Reviewer #1: N/A

Reviewer #2: Yes

3. Have the authors made all data underlying the findings in their manuscript fully available?

Reviewer #1: Yes

Reviewer #2: Yes

4. Is the manuscript presented in an intelligible fashion and written in standard English?

Reviewer #1: No

Reviewer #2: Yes

5. Review Comments to the Author

Reviewer #1: Recommendation:

Major revision

Comments:

- In the molecular docking section explain why PDB ID: 6IZZ was used to perform docking studies. Describe the mutations, missing regions and active or inactive states of the receptors.

- The binding interactions between Catechin and RdRp protease should be visualized in 2D and 3D images.

- The conclusion part should be more informative.

- The word in vivo and in vitro should be written in Italic form throughout the manuscript.

- The word IC50 should be corrected to IC50.

- Many paragraphs need to be justified.

- In the introduction section, the authors need to discuss some recent reports about molecular docking and its role in the interactions of potential candidates with their target. You can depend on the following articles

- https://doi.org/10.3390/molecules27227719

- https://doi.org/10.3390/molecules27185859

- https://doi.org/10.3390/molecules27155047

- The quality of the images of MD simulation are very low and should be improved.

- The manuscript should be written in standard English

- Writing of the manuscript for language and grammar needs to be thoroughly checked.

Reviewer #2: Dear Authors,

First:

I have reviewed the section of your manuscript concerning the Molecular Dynamic (MD) simulations and would like to discuss the interpretation of the RMSD (Root Mean Square Deviation) and RMSF (Root Mean Square Fluctuation) values in relation to the stability of the (+)-Catechin-protein complex.

In your manuscript, you mention that the (+)-Catechin-protein complex exhibited stability throughout the 100 ns MD simulation period. However, I observed that the RMSD value for the complex increased significantly, reaching up to 25.5 Å, and the average RMSD was reported as 18.0 Å【17†source】. Typically, a lower RMSD value during a simulation indicates a stable protein-ligand complex, while a higher RMSD value suggests less stability.

Furthermore, the RMSF values, which reflect the flexibility and mobility of amino acid residues during the simulation, also showed considerable variation, with the average RMSF observed for RdRp following binding of (+)-Catechin being 10.0 Å【18†source】. Higher RMSF values typically indicate greater flexibility and potentially less stability in specific regions of the protein.

Given these observations, it seems that there is a contradiction in stating that the complex remains stable while the RMSD and RMSF values suggest significant fluctuations and changes in the structure. It would be beneficial for the manuscript if you could clarify the following points:

1. Interpretation of RMSD Values: How do you reconcile the relatively high RMSD values with the conclusion that the complex is stable? Is there a specific criterion or threshold you are using to define stability in this context?

2. RMSF Insights: Could you provide more insights into the high RMSF values observed? Are these fluctuations concentrated in specific regions of the protein, and do they affect the overall stability of the complex?

3. Impact on Binding Efficacy: How do these fluctuations influence the binding efficacy and potential inhibitory action of (+)-Catechin on the RdRp protease?

Your clarification on these points would greatly enhance the understanding of the MD simulation results and their implications for the stability and efficacy of the (+)-Catechin-protein complex as a potential DENV inhibitor.

Second:

I have carefully reviewed your manuscript, particularly the Molecular Dynamic (MD) simulations section, focusing on the RMSF (Root Mean Square Fluctuation) values of the (+)-Catechin-protein complex. One key aspect that appears to be missing in your analysis is the comparison of these RMSF values with the RMSF of the apo protein (the unbound state of the protein).

In your study, you report the RMSF values for the (+)-Catechin-protein complex, highlighting the flexibility and mobility of amino acid residues during the simulation. These values provide essential insights into the stability and conformational dynamics of the protein-ligand complex【18†source】. However, for a comprehensive understanding of how the binding of (+)-Catechin affects the protein, it would be beneficial to compare these results with the RMSF of the protein in its unbound state. Such a comparison can offer valuable information on the following:

1. Effect of Ligand Binding on Protein Dynamics**: Comparing the RMSF values of the apo protein with the complex can reveal if and how the binding of (+)-Catechin alters the dynamic behavior of the protein. This can be particularly informative for understanding any conformational changes or stabilization effects induced by ligand binding.

2.Regions of Increased or Decreased Flexibility**: By contrasting the RMSF profiles of the apo and bound states, you can identify specific regions in the protein that become more or less flexible upon ligand binding. This can provide insights into the mechanism of inhibition and the structural basis of the protein's functional modulation.

3. Benchmark for Stability Analysis**: Comparing the RMSF values of the apo and bound states can serve as a benchmark to assess the relative stability imparted by ligand binding. This comparison could strengthen your conclusions about the stability and efficacy of the (+)-Catechin-protein complex as a potential inhibitor.

I believe that incorporating this comparative analysis would greatly enhance the depth and significance of your MD simulation results, offering a more nuanced understanding of the interaction dynamics between (+)-Catechin and the RdRp protease.

Third:

Regarding the section "3.5. Density Function Theory (DFT) Calculations" in your manuscript, I would like to offer a brief comment for improvement. While the results in Table 1 provide valuable data on the electronic properties of (+)-Catechin, there appears to be a missed opportunity in not further interpreting these results to deduce the nature of this compound in terms of its reactivity and interaction mechanism.

Your analysis effectively identifies the HOMO and LUMO energy levels, and the energy gap, but stops short of extrapolating how these properties translate into the compound's behavior as an electron donor or acceptor. Such an interpretation is crucial for understanding the molecular basis of its potential inhibitory action on the RdRp protease.

This additional step of analysis could significantly strengthen your DFT section, providing deeper insights into the electronic characteristics and reactivity of (+)-Catechin, and how these properties contribute to its function as a potential inhibitor. It would be beneficial to readers and the field if this aspect could be explored and discussed.

fourth:

I have reviewed your manuscript and would like to recommend the inclusion of some recent and relevant references that could significantly enhance the depth and context of your work. These references provide insights and data that align closely with your study's focus and could offer additional perspectives or comparative data to strengthen your arguments.

1. [Molecules: DOI 10.3390/molecules27196320]

2. Reference on Molecular Docking and Simulation Techniques:

[Crystals: DOI 10.3390/cryst13071086](https://doi.org/10.3390/cryst13071086)

3. Reference on Density Functional Theory (DFT) Analysis:

[Crystals: DOI 10.3390/cryst13071020]

This article covers the latest developments in DFT calculations, potentially providing a broader context or newer techniques that could be applied to your DFT analysis.

6. PLOS authors have the option to publish the peer review history of their article (what does this mean?). If published, this will include your full peer review and any attached files.

Reviewer #1: No

Reviewer #2: No

---

## [Author Response · Author response to Decision Letter 0]

14 Jan 2024

Dear Editors and Reviewers,

Thank you for your letter and the detailed comments provided by the reviewers regarding our manuscript titled "Phenolic Compounds of Theobroma cacao L. show potential against dengue RdRp Protease enzyme inhibition by In-silico Docking, DFT study, MD simulation, and MMGBSA and DFT calculation" (Manuscript ID PONE-D-23-29095). We appreciate the time and effort invested in evaluating our work.

We have carefully addressed each comment and incorporated the necessary revisions into the manuscript. The changes have been highlighted using the track changes feature, making it easier for you and the reviewers to identify the modifications made. We believe that these revisions significantly addressed the concerns raised during the review process and enhanced the clarity and rigor of our research.

We sincerely value the Editors' and Reviewers' contributions to the improvement of our manuscript. We are confident that the revisions have strengthened the scientific merit of our work, and we believe that it is now suitable for publication in your journal.

We are at your disposal for any further questions or clarifications. Your guidance is highly appreciated, and we look forward to the possibility of having our work published in your esteemed journal.

Thank you for your time and consideration. Once again, thank you very much for your comments and suggestions.

Best regards,

Md. Rabiul Islam, PhD

Reviewer #1

Comment: In the molecular docking section explain why PDB ID: 6IZZ was used to perform docking studies. Describe the mutations, missing regions and active or inactive states of the receptors.

Response: According to the original study of 6IZZ protein’s crystal structure, it is reported that the amino acid residue Cys highly conserved in DENV 1-4 serotypes and plays very important role for the enzymatic activity. Mutation of this residue with Asn and Glu altered the efficacy of the inhibitor by increased IC50 value of the inhibitor but not diminishing the activity. More concisely, Cys 780 and Cys709 amino acid residue at the binding site critical for enzymatic activity. So, inhibitors binding to Cys780 and Cys709 are expected to inhibit the RdRp activity (Shimizu et al., 2019).

Moreover, The active binding site(s) of 6IZZ have been revealed from the crystal structure of this protein where the co-crystalized ligand (Inhibitor) bound and the ligand-amino acid interactions are explained. So, any missing residue other than active binding site may not influence the activity of the inhibitors. In our study, catechin bound to the amino acids which are located at the binding site.

Regarding the active and inactive state of DENV3 NS65 RdRp, not much conformational state is described in the literature. However, generally, the functional mechanism of favivirus RdRp remains in a resting mode in the closed conformation as seen in the crystal structures with the priming loop in an extended conformation. The NS5 protein undergoes a large conformational change with a concomitant retraction of the priming loop leading to an opening of the RNA exit tunnel. This conformational change is thought to be the rate-limiting step for the RNA polymerase activity (Choi et al. 2012). Finally, the protein is locked in an open conformation, allowing processive RNA polymerization.

Reference: 

• Shimizu H, Saito A, Mikuni J, Nakayama EE, Koyama H, Honma T, Shirouzu M, Sekine SI, Shioda T. Discovery of a small molecule inhibitor targeting dengue virus NS5 RNA-dependent RNA polymerase. PLoS Negl Trop Dis. 2019 Nov 18;13(11):e0007894. doi: 10.1371/journal.pntd.0007894. PMID: 31738758; PMCID: PMC6886872.

• Choi K.H. Viral polymerases. Adv. Exp. Med. Biol. 2012;726:267–304.

Comment: The binding interactions between Catechin and RdRp protease should be visualized in 2D and 3D images.

Response: Thanks for the suggestion. We have added the images as highlighted in yellow in the manuscript. Please see page 12.

Comment: The conclusion part should be more informative.

Response: We have improved the conclusion

Comment: The word in vivo and in vitro should be written in Italic form throughout the manuscript.

Response: We have corrected accordingly.

Comment: The word IC50 should be corrected to IC50

Response: We have corrected accordingly. Please find the correction on page 7, Line 193.

Comment: Many paragraphs need to be justified.

Response: We have corrected accordingly.

Comment: In the introduction section, the authors need to discuss some recent reports about molecular docking and its role in the interactions of potential candidates with their target. You can depend on the following articles

- https://doi.org/10.3390/molecules27227719

- https://doi.org/10.3390/molecules27185859

- https://doi.org/10.3390/molecules27155047

Response: We thank the reviewer for the suggestion. We have added a paragraph with proper citations in the Introduction section which is highlighted with gray color. Please see Page 5, Line 132-147.

Comment: The quality of the images of MD simulation is very low and should be improved.

Response: Thanks for the suggestion. We have added new images as per your suggestion.

Comment: The manuscript should be written in standard English

Response: We have improved the language.

Comment: Writing of the manuscript for language and grammar needs to be thoroughly checked.

Response: Grammatical errors have been checked and resolved by using Grammarly tool.

Reviewer #2

Comments: First:

I have reviewed the section of your manuscript concerning the Molecular Dynamic (MD) simulations and would like to discuss the interpretation of the RMSD (Root Mean Square Deviation) and RMSF (Root Mean Square Fluctuation) values in relation to the stability of the (+)-Catechin-protein complex.

In your manuscript, you mention that the (+)-Catechin-protein complex exhibited stability throughout the 100 ns MD simulation period. However, I observed that the RMSD value for the complex increased significantly, reaching up to 25.5 Å, and the average RMSD was reported as 18.0 Å【17†source】. Typically, a lower RMSD value during a simulation indicates a stable protein-ligand complex, while a higher RMSD value suggests less stability.

Furthermore, the RMSF values, which reflect the flexibility and mobility of amino acid residues during the simulation, also showed considerable variation, with the average RMSF observed for RdRp following binding of (+)-Catechin being 10.0 Å【18†source】. Higher RMSF values typically indicate greater flexibility and potentially less stability in specific regions of the protein.

Given these observations, it seems that there is a contradiction in stating that the complex remains stable while the RMSD and RMSF values suggest significant fluctuations and changes in the structure. It would be beneficial for the manuscript if you could clarify the following points:

1. Interpretation of RMSD Values: How do you reconcile the relatively high RMSD values with the conclusion that the complex is stable? Is there a specific criterion or threshold you are using to define stability in this context?

2. RMSF Insights: Could you provide more insights into the high RMSF values observed? Are these fluctuations concentrated in specific regions of the protein, and do they affect the overall stability of the complex?

3. Impact on Binding Efficacy: How do these fluctuations influence the binding efficacy and potential inhibitory action of (+)-Catechin on the RdRp protease?

Your clarification on these points would greatly enhance the understanding of the MD simulation results and their implications for the stability and efficacy of the (+)-Catechin-protein complex as a potential DENV inhibitor.

Response: The authors would like to thank the reviewer for his deep and insightful commentary on MD simulation study. It is really appreciable that the reviewer has made detail explanations on is comments and provided very useful suggestions to improve the manuscript. 

To answer the on the high RMSD value than the protein's RMSD, it is indicative that the ligand shifted from its initial binding position to occupy a different binding position. This result is in accordance with the first and last pose data for ligand. The RMSD until around 25ns was lower. Although there was a sudden jump at around 38-40 ns but eventually it dropped immediately and attained a relatively high but steady state until 100 ns without much fluctuation in the RMSD value. The increased RMSD suggests change in conformation, and there are possibilities that the ligand shifted to a new binding location that was different from its initial binding location and achieve stability (Alandijany et al., 2023).

Similarly, a relatively higher ligand RMSF plot indicates that that the ligand was constantly changing its binding pose at the in search of a more stable binding pose (Kakhar et al., 2023).

References:

Alandijany TA, El-Daly MM, Tolah AM, Bajrai LH, Khateb AM, Alsaady IM, Altwaim SA, Dubey A, Dwivedi VD, Azhar EI. Investigating the Mechanism of Action of Anti-Dengue Compounds as Potential Binders of Zika Virus RNA-Dependent RNA Polymerase. Viruses. 2023 Jul 4;15(7):1501. doi: 10.3390/v15071501. PMID: 37515188; PMCID: PMC10384299.

Kakhar Umar A, Zothantluanga JH, Luckanagul JA, Limpikirati P, Sriwidodo S. Structure-based computational screening of 470 natural quercetin derivatives for identification of SARS-CoV-2 Mpro inhibitor. PeerJ. 2023 Mar 14;11:e14915. doi: 10.7717/peerj.14915. PMID: 36935912; PMCID: PMC10022500.

Comment: Second:

I have carefully reviewed your manuscript, particularly the Molecular Dynamic (MD) simulations section, focusing on the RMSF (Root Mean Square Fluctuation) values of the (+)-Catechin-protein complex. One key aspect that appears to be missing in your analysis is the comparison of these RMSF values with the RMSF of the apo protein (the unbound state of the protein).

In your study, you report the RMSF values for the (+)-Catechin-protein complex, highlighting the flexibility and mobility of amino acid residues during the simulation. These values provide essential insights into the stability and conformational dynamics of the protein-ligand complex【18†source】. However, for a comprehensive understanding of how the binding of (+)-Catechin affects the protein, it would be beneficial to compare these results with the RMSF of the protein in its unbound state. Such a comparison can offer valuable information on the following:

1. Effect of Ligand Binding on Protein Dynamics**: Comparing the RMSF values of the apo protein with the complex can reveal if and how the binding of (+)-Catechin alters the dynamic behavior of the protein. This can be particularly informative for understanding any conformational changes or stabilization effects induced by ligand binding.

2.Regions of Increased or Decreased Flexibility**: By contrasting the RMSF profiles of the apo and bound states, you can identify specific regions in the protein that become more or less flexible upon ligand binding. This can provide insights into the mechanism of inhibition and the structural basis of the protein's functional modulation.

3. Benchmark for Stability Analysis**: Comparing the RMSF values of the apo and bound states can serve as a benchmark to assess the relative stability imparted by ligand binding. This comparison could strengthen your conclusions about the stability and efficacy of the (+)-Catechin-protein complex as a potential inhibitor.

I believe that incorporating this comparative analysis would greatly enhance the depth and significance of your MD simulation results, offering a more nuanced understanding of the interaction dynamics between (+)-Catechin and the RdRp protease.

Response: Again we would like to thank the reviewer for the valuable remarks and suggestion on MD simulation. In the first version of our manuscript the comparative result for the preference compound (Panduratin A) was not discussed because we did not performed MD simulation for the reference compound. In order to have a better and comprehensive understanding of the potentiality of our suggested best compound (catechin), We performed the whole MD simulation study again. However, this time we were not able to access the Desmond/ schrodinger suite due to expiration of License. Thus we sought collaboration with another colleague who performed the MD simulation using GROMACS. The whole study result is added in the manuscript in both methodology and Result and Discussion sections which we hope will give better insights on our findings. Both RMSD and RMSF results were compared with Apo protein and the reference compound. Please refer to page 25, section 3.8.

Comments: Third:

Regarding the section "3.5. Density Function Theory (DFT) Calculations" in your manuscript, I would like to offer a brief comment for improvement. While the results in Table 1 provide valuable data on the electronic properties of (+)-Catechin, there appears to be a missed opportunity in not further interpreting these results to deduce the nature of this compound in terms of its reactivity and interaction mechanism.

Your analysis effectively identifies the HOMO and LUMO energy levels, and the energy gap, but stops short of extrapolating how these properties translate into the compound's behavior as an electron donor or acceptor. Such an interpretation is crucial for understanding the molecular basis of its potential inhibitory action on the RdRp protease.

This additional step of analysis could significantly strengthen your DFT section, providing deeper insights into the electronic characteristics and reactivity of (+)-Catechin, and how these properties contribute to its function as a potential inhibitor. It would be beneficial to readers and the field if this aspect could be explored and discussed.

Response: Once again, thanks to the reviewer for the insightful comment on DFT study. We have incorporated the following discussion in the manuscript in the result section which is highlighted in green in Pages 17-19

“The comparison between catechin and anduratin A highlights distinct reactivity and interaction mechanisms based on various molecular properties. Panduratin A exhibits considerably higher total energy and binding energy in comparison to catechin, indicating slightly lower stability and a potentiality for forming robust interactions during reactions or when binding to other molecules as compared to panduratin A. Moreover, Panduratin A possesses a substantially larger dipole moment, signifying higher polarity and a heightened potential for strong electrostatic interactions. While both compounds share similar HOMO energies, catechin showcases a slightly higher LUMO energy and a extended band energy gap, hinting at a slightly lower reactivity potential due to its lesser likelihood to donate or accept electrons. Additionally, catechin's increased hardness and lower softness compared to panduratin A suggest a more controlled reactivity, indicating a lesser susceptibility to changes in electron density. The compound's higher electronegativity and electrophilicity further reinforce its potential for increased reactivity and stronger interactions in various chemical contexts. It's crucial to note that these observations offer a molecular perspective, and the actual reactivity and interaction mechanisms can be influenced by external factors and specific experimental conditions.

Understanding the molecular properties of compounds such as Catechin and Panduratin A is critical when exploring their potential inhibitory action on the RdRp (RNA-dependent RNA polymerase) enzyme in diseases like dengue, where RdRp plays a pivotal role in the virus's replication. Panduratin A's traits, like higher stability indicated by elevated total and binding energies, suggest its capacity for establishing robust interactions with the RdRp enzyme. The larger dipole moment and increased polarity might enable Panduratin A to engage in specific interactions with crucial regions of the RdRp protein, potentially influencing its binding capacity and inhibitory potential. Comparable and similar properties of catechin may also show favorable inhibitory action. Additionally, the lower LUMO energy and smaller band gap energy of compounds hint at a greater likelihood for electron transfer, advantageous in disrupting electron transfer processes or active sites of the RdRp enzyme, hindering its function and impeding viral replication. Higher hardness and lower softness imply controlled reactivity, beneficial for stable and specific interactions with active sites or critical residues within the RdRp enzyme, potentially interfering with its catalytic activity and inhibiting viral replication. Moreover, compound’s higher electronegativity and electrophilicity suggest its capability to act as an electron acceptor, forming robust interactions with specific functional groups or residues within the RdRp enzyme, possibly disrupting its function and contributing to inhibitory action against the dengue virus. Despite the fact that, catechin showed slightly lower outcomes in DFT result in contrast to panduran A, its potentiality to interact with the RdRp cant not be denied. While these molecular properties provide insights into both panduratin A and catechin's potential inhibitory action on the RdRp enzyme in dengue virus replication, rigorous experimental investigations, including enzymatic assays, and in vitro studies, are necessary to validate these interactions and elucidate specific mechanisms by which catechin interacts with the RdRp protease, thus determining its effectiveness as a potential therapeutic agent against dengue virus infections.

The molecular properties exhibited by compounds like catechin play a critical role in determining their potential as inhibitors, particularly concerning their interaction with specific enzymes like the RdRp protease. These properties contribute to its function as a potential inhibitor in several ways: i) Stability and Binding Energy: A high total and binding energies suggest that catechin could form more stable interactions with the active sites or binding pockets of the RdRp enzyme. This stability is crucial for a strong and long-lasting inhibitor-enzyme interaction, potentially disrupting the enzyme's function necessary for viral replication; ii) Dipole moment and polarity: Larger dipole moment and increased polarity allow it to interact with the RdRp enzyme via electrostatic forces. These interactions may facilitate the compound's binding to specific regions of the enzyme, enhancing its inhibitory potential by disrupting the enzyme's active sites or altering its conformation; iii) HOMO and LUMO energy, band gap energy: Lower LUMO energy and smaller band gap energy suggest its propensity for electron transfer. This characteristic is beneficial as it may interfere with the electron transfer processes essential for the RdRp enzyme's catalytic activity, thereby inhibiting the enzyme's function and viral replication; iv). Hardness and softness: Higher hardness and lower softness imply controlled reactivity. This controlled reactivity could enable the compound to form stable and specific interactions with the active sites or critical residues within the RdRp enzyme, effectively disrupting its function and inhibiting viral replication; v) Electronegativity and electrophilicity: Higher electronegativity and electrophilicity indicate potential to accept electrons and form strong interactions with specific functional groups or residues within the RdRp enzyme. These interactions could hinder the enzyme's function and contribute to its inhibition.

Collectively, these molecular properties of catechin which is comparable with panduratin A in this study may contribute to its potential as an inhibitor by facilitating strong and specific interactions with the RdRp enzyme. By binding to critical sites or interfering with essential processes in the enzyme's function, catechin in a similar fashion to panduratin A may disrupt the replication cycle of the virus, making it a promising candidate for further exploration as a potential therapeutic agent against viral infections, such as dengue.”

Comments: fourth:

I have reviewed your manuscript and would like to recommend the inclusion of some recent and relevant references that could significantly enhance the depth and context of your work. These references provide insights and data that align closely with your study's focus and could offer additional perspectives or comparative data to strengthen your arguments.

1. [Molecules: DOI 10.3390/molecules27196320]

2. Reference on Molecular Docking and Simulation Techniques:

[Crystals: DOI 10.3390/cryst13071086](https://doi.org/10.3390/cryst13071086)

3. Reference on Density Functional Theory (DFT) Analysis:

[Crystals: DOI 10.3390/cryst13071020]

This article covers the latest developments in DFT calculations, potentially providing a broader context or newer techniques that could be applied to your DFT analysis.

Response: Thank you for referring some of the useful manuscripts. We have reviewed then and cited a where necessary. It can be seen on page 27, citation number [57].

---

## [Decision Letter · Decision Letter 1]

6 Feb 2024

Phenolic Compounds of Theobroma cacao L. show potential against dengue RdRp Protease enzyme inhibition by In-silico Docking, DFT study, MD simulation and MMGBSA calculation

PONE-D-23-34561R1

Dear Dr. Islam,

We’re pleased to inform you that your manuscript has been judged scientifically suitable for publication and will be formally accepted for publication once it meets all outstanding technical requirements.

Kind regards,

Erman Salih Istifli, PhD

Academic Editor

PLOS ONE

Additional Editor Comments (optional):

I have assessed the manuscript and despite the conflicting reviewers recommendations, I feel this study is thorough and the manuscript is suitable for publication. The study by Islam et al. employed sufficient methodology required for a structure-based drug design study.

Reviewers' comments:

Reviewer's Responses to Questions

**Comments to the Author**

1. If the authors have adequately addressed your comments raised in a previous round of review and you feel that this manuscript is now acceptable for publication, you may indicate that here to bypass the “Comments to the Author” section, enter your conflict of interest statement in the “Confidential to Editor” section, and submit your "Accept" recommendation.

Reviewer #1: All comments have been addressed

Reviewer #2: All comments have been addressed

2. Is the manuscript technically sound, and do the data support the conclusions?

Reviewer #1: Yes

Reviewer #2: No

3. Has the statistical analysis been performed appropriately and rigorously? 

Reviewer #1: N/A

Reviewer #2: I Don't Know

4. Have the authors made all data underlying the findings in their manuscript fully available?

Reviewer #1: Yes

Reviewer #2: Yes

5. Is the manuscript presented in an intelligible fashion and written in standard English?

Reviewer #1: Yes

Reviewer #2: Yes

6. Review Comments to the Author

Reviewer #1: The authors have refuted most of my comments and therefore I recommend accepting the manuscript in the current form

Reviewer #2: Title: Disapproval of Manuscript Review Response for the Study on Phenolic Compounds of Theobroma cacao L. in "PLOS ONE"

Dear Editors of PLOS ONE,

I am writing to express my dissatisfaction with the responses provided to comments raised during the review of the manuscript titled "Phenolic Compounds of Theobroma cacao L. show potential against dengue RdRp Protease enzyme inhibition by In-silico Docking, DFT study, MD simulation and MMGBSA calculation."

Despite having carefully reviewed the study, I found the responses to the comments to be unsatisfactory. The explanations provided were not sufficiently scientific, and the details were not adequately addressed. I believe that the manuscript, in its current state, is not suitable for publication. The lack of precision and scientific rigor in the explanations raises concerns about the credibility and reliability of the presented findings.

Therefore, I must express my opinion that, from my perspective, the manuscript is not suitable for publication in its current form and should be rejected. I hope that my concerns are taken into consideration during the evaluation process.

Thank you for your attention to this matter.

7. PLOS authors have the option to publish the peer review history of their article (what does this mean?). If published, this will include your full peer review and any attached files.

Reviewer #1: No

Reviewer #2: No

---

## [Editor Report · Acceptance letter]

4 Mar 2024

PONE-D-23-34561R1 

PLOS ONE

Dear Dr. Islam, 

I'm pleased to inform you that your manuscript has been deemed suitable for publication in PLOS ONE. Congratulations! Your manuscript is now being handed over to our production team.

Kind regards, 

on behalf of

Assoc. Prof. Dr. Erman Salih Istifli 

Academic Editor

PLOS ONE